# Evaluation of Scar Quality after Treatment of Superficial Burns with Dressilk^®^ and Suprathel^®^—In an Intraindividual Clinical Setting

**DOI:** 10.3390/jcm11102857

**Published:** 2022-05-18

**Authors:** Jennifer Lynn Schiefer, Janine Andreae, Paul Christian Fuchs, Rolf Lefering, Paul Immanuel Heidekrueger, Alexandra Schulz, Mahsa Bagheri

**Affiliations:** 1Clinic of Plastic, Reconstructive, Hand and Burn Surgery, Hospital Cologne Merheim, University of Witten-Herdecke, 58448 Witten, Germany; smilingjanine@googlemail.com (J.A.); fuchsp@kliniken-koeln.de (P.C.F.); alexandra_schulz@hotmail.com (A.S.); bagherim@kliniken-koeln.de (M.B.); 2Institute for Research in Operative Medicine (IFOM), University of Witten-Herdecke, 50933 Cologne, Germany; rolf.lefering@uni-wh.de; 3Center of Plastic, Aesthetic, Hand and Reconstructive Surgery, University of Regensburg, 93053 Regensburg, Germany; paul@heidekrueger.net

**Keywords:** superficial burns, silk, Suprathel, Dressilk, wound healing

## Abstract

Background: Various synthetic and biological wound dressings are available for the treatment of superficial burns, and standard care differs among hospitals. Nevertheless, the search for an ideal wound dressing offering a safe healing environment as well as optimal scar quality while being economically attractive is a continuing process. In recent years, Dressilk^®^, which consists of pure silk, has become the standard of care for the treatment of superficial burns in our hospital. However, no long-term scar-evaluation studies have been performed to compare Dressilk^®^ with the often-used and more expensive Suprathel^®^ in the treatment of superficial burns. Methods: Subjective and objective scar evaluations were performed three, six, and twelve months after treatment in patients who received simultaneous treatment of 20 superficial burn wounds with both Suprathel^®^ and Dressilk^®^. The evaluations were performed using the Vancouver Scar Scale, the Cutometer^®^, Mexameter^®^, Tewameter^®^, and the O2C^®^. Results: Both dressings showed mostly equivalent results in subjective scar evaluations. In the objective scar evaluations, the wounds treated with Dressilk^®^ showed a faster return to the qualities of non-injured skin. Wound areas treated with the two dressings showed no significant differences in elasticity and transepidermal water loss after 12 months. Only oxygen saturation was significantly lower in wound areas treated with Suprathel^®^ (*p* = 0.008). Subjectively, wound areas treated with Dressilk^®^ showed significantly higher pigmentation after six months, which was not apparent after 12 months. Conclusion: Both wound dressings led to esthetically satisfying scar recovery without significant differences from normal uninjured skin after 12 months. Therefore, Dressilk^®^ remains an economically and clinically interesting alternative to Suprathel^®^ for the treatment of superficial burns.

## 1. Introduction

The appearance and quality of burn scars are of major importance for the affected patients [1]. With advancements in medical treatments and reductions in mortality rates after burn injuries [2], the long-term consequences of scar quality, such as textural or pigmentation problems, are now receiving more attention from surgeons. Thus, an optimal wound dressing that can minimize patient requests to improve scar quality is needed [1]. To this end, objective and reproducible scar evaluations are necessary to measure not only the quality of the different treatments, but also the efficacy of anti-scar techniques [3]. 

Currently, numerous wound dressings are available for the treatment of superficial burns [4,5,6,7]. In particular, synthetic and commercially available materials that can accelerate wound healing and reduce scarring are the main focus of current research in burn medicine [6,8,9,10,11,12,13]. Different materials have been established for the treatment of burns in recent years. Suprathel^®^, a biosynthetic copolymer wound dressing, was originally developed to improve wound closure, especially of split-skin graft donor sites, and in the conservative treatment of second-degree burns. In recent studies, it was shown to promote wound healing and reduce wound infection [14]. As an alternative material, silk spun by silkworm consists of the protein fibroin and shows many promising characteristics such as biocompatibility, tunable mechanical properties, minimal inflammation in host tissue, low cost and ease of use [6,15]. Due to these qualities, Dressilk^®^ has become the standard of care (SOC) for the treatment of superficial partial-thickness burn wounds at our burn center in recent years. 

Since many burn centers prefer using Suprathel^®^ for these wounds, we had previously compared wound healing and patient satisfaction after the treatment of superficial burns with Dressilk^®^ and Suprathel^®^ [16]. However, to our knowledge, no study has attempted a direct comparison of scarring after burn wound treatment with Suprathel^®^ and Dressilk^®^. Therefore, we aimed to subjectively and objectively evaluate long-term scar quality after wound treatment with both wound dressings.

## 2. Materials and Methods

Study design: The present study evaluated the scar quality of superficial partial-thickness burn wounds after simultaneous treatment with Suprathel^®^ and Dressilk^®^. The study protocol had been reviewed and approved in advance by the Ethical Review Committee of the University of Witten Herdecke, Germany (number 5/2017), and the experimental procedures were conducted in accordance with the Declaration of Helsinki. Complete informed consent was obtained from all patients. 

A total of 20 patients with superficial partial-thickness burns had previously been treated with Suprathel^®^ and Dressilk^®^ in an intra-individual study design, with half of the burned areas receiving Suprathel^®^ and the other half being treated with Dressilk^®^ (Figure 1) [16]. All of the treated patients were at least 18 years old, had a superficial partial-thickness burn caused by contact with a hot surface, flames, or a hot liquid and a wound area ≥0.3% of the total burned surface area (TBSA). For this study, all 20 patients were invited for follow-up examinations to evaluate scar quality after three, six, and twelve months (Figure 2). In the follow-up assessments, all study areas were photo-documented by standardized means and the scars assessed subjectively and objectively. We excluded patients who did not provide consent for the follow-up assessments and those who did not comply with the protocol for the examinations.

**Subjective scar assessment** was performed using the Vancouver Scar Scale (VSS). The traditional VSS is a validated subjective scale for scar assessment that has been described in detail in previous studies [17,18,19,20]. 

**Objective scar assessment** was performed with established objective scar-assessment tools such as the Cutometer^®^, Mexameter^®^, Tewameter^®^ (Courage + Khazala Electronic GmbH, Cologne, Germany), and the O2C^®^ (Oxygen to see; LEA Medizintechnik, Giessen, Germany). More details regarding these tools are provided in the following paragraphs: 

The **Cutometer^®^** is used to determine the viscoelasticity and pliability of skin. The measurement principle of this tool is based on the concept of negative pressure, which is produced with a pump in the device and draws the skin into the opening of the measuring probe [21]. Three different measurement variables were taken into account for this study: (1) R0, the maximum suction depth of the skin, which represents the firmness of the skin; (2) R2, the difference between suction and retraction, i.e., elasticity, and (3) F1, the area under the curve up to the maximum amplitude [21].

The **Mexameter^®^** is used to assess skin discoloration, which is evaluated measuring vascularization (erythema) and pigmentation (Melanin). The device emits light of three different wavelengths and then evaluates the light reflected by the skin in order to calculate the amount of light absorbed [22].

The **Tewameter^®^** is a modern digital evaporimeter that non-invasively measures the evaporative water loss (EWL) of the skin. The EWL is defined as the quantity of water in grams that passes through the skin to the surrounding atmosphere per hour and area (m^2^) [23]. Two sensors for temperature and relative humidity measure the EWL indirectly with the help of a formula [24]. The transepidermal water loss (TEWL), an essential parameter indicating the efficiency of the skin barrier, and the skin surface water loss (SSWL), which indicates the water-binding capacity, were evaluated [25].

**The O2C^®^** assesses perfusion by combining photo-spectroscopy and Laser Doppler to evaluate superficial oxygen saturation (sO_2_), hemoglobin concentration (rHb), and flow [26,27].

Statistical analysis: We used Microsoft Excel (2017, Microsoft, USA) to manage data and create the charts. The data evaluated in this study were collected prospectively. After a thorough review of the data, SPSS (Version 21, IBM, USA) was used for final statistical analysis. Statistical significance was considered at *p* ≤ 0.05. The Friedman and Wilcoxon test was performed to identify statistically significant differences between the subgroups.

## 3. Results

Twenty patients (12 males and 8 females) were treated with both wound dressings between May 2017 and May 2018 and asked to participate in this follow-up study for scar evaluation. The patient characteristics are presented in Table 1.

### 3.1. Subjective Scar Evaluation

The VSS evaluations showed no significant differences in blood circulation, pigmentation, elasticity and skin thickness between the wound areas treated after 12 months. A significant difference in blood flow (*p* = 0.005) was only observed after six months, when the burn wound treated with Dressilk^®^ was less red and significantly more pigmented (*p* = 0.02) than the second-degree burn treated with Suprathel^®^. 

### 3.2. Objective Scar Evaluations

Subsequently, scarring was objectively evaluated in comparison with the selected healthy skin areas in the follow-up examinations performed after three, six, and twelve months and compared to the selected healthy skin areas.

*Cutometer^®^*—In assessments of the firmness of the skin (R_0_ value), significant differences could be seen between the two dressings after three months (*p* = 0.013). Suprathel^®^ showed a significant difference in comparison with the healthy skin (*p* = 0.024). At this point, the elasticity (R_2_ value) of the wound treated with Suprathel^®^ (*p* = 0.017) differed significantly from the standard value (Figure 3). 

However, the R_2_ values for both materials were similar (*p* = 0.057). For the F1 value, a significant difference between the two dressings could only be detected after three months (*p* = 0.01). After six and twelve months, no further statistical differences were detected in the measured values (Table 2).

*Mexameter^®^*—No significant differences were observed in pigmentation between the dressings after three and six months. 

A significant difference in comparison with the normal skin could be detected at six months after Suprathel^®^ treatment (*p* = 0.021). After 12 months, a significant difference in pigmentation (*p* = 0.009) could be observed between both dressings (Figure 4). 

Thus, the wound areas treated with Dressilk^®^ showed no significant difference in pigmentation in comparison with normal skin, whereas those treated with Suprathel^®^ showed a significant difference (*p* = 0.004). In assessments of erythema, both dressings showed significantly higher values than those for normal skin after three, six, and twelve months, whereas the inter-therapeutic difference was not significant (Table 3).

*Tewameter^®^*—The transepidermal water loss (TEWL) values for the wounds treated with Dressilk^®^ did not differ significantly from those for the wounds treated with Suprathel^®^ after months three and twelve (see Table 4). During both measurements, the TEWL was significantly lower than the values for healthy skin.

Interestingly, a significant difference was observed in the TEWL values between the two wound dressings after 6 months (*p* = 0.017; Figure 5), with the wounds treated with Dressilk^®^ showing no significant difference to normal skin, unlike Suprathel^®^ (*p* = 0.044).

*Oxygen to see (O2C)^®^*—After three months, there were no significant differences between the two dressings in terms of sO_2_ and rHb values. However, the blood flow rate differed at this point and was significantly higher in the wound areas treated with Suprathel^®^ (*p* = 0.04). In comparison with normal skin, the wound areas treated with both dressings showed significantly higher values for all measured parameters after 3 months (Figure 6).

In assessments of sO_2_, the wounds treated with Dressilk^®^ did not differ from the healthy skin after month six (*p* = 0.053). However, after 12 months, the sO_2_ values of the differently treated wounds differed significantly (*p* = 0.004), and the sO_2_ of the burn wounds treated with Dressilk^®^ was significantly lower. Similar to the findings after month six, the wound treated with Dressilk^®^ no longer differed from normal skin, in contrast to the wound treated with Suprathel^®^ (*p* = 0.008). The rHb and flow rate showed no significant difference between the therapies at this point. However, Suprathel^®^ (*p* = 0.002) and Dressilk^®^ (*p* = 0.007) still showed significant differences in rHb in comparison with normal skin, whereas the flow rates in wounds treated with both dressings (*p* > 0.05) were consistent with that of normal skin (Table 5). Altogether, after 12 months, significant differences between the areas treated with the two dressings were solely detected with the O2C regarding the sO_2_ value. All other values showed no significant difference between the two treatments at this point.

Overall, after 12 months, only a few differences between the two differently treated areas could be detected. The VSS, Cutometer and Tewameter showed no differences; only the melanin values from the Mexameter and the sO_2_ value from the O2C were significantly different.

## 4. Discussion

To the best of our knowledge, this is the first study using standardized and reproducible scar-evaluation tools to compare the scar quality after treatment with the wound dressings Dressilk^®^ and Suprathel^®^.

### 4.1. Scar Assessment

#### 4.1.1. Subjective Scar Assessment

In burn rehabilitation, patient satisfaction with the appearance of their scar is of high importance [28]. The outcome of the treatment procedure is heavily influenced by the patients’ opinion about the scarring [29,30]. With a focus on patient satisfaction, we performed VSS assessments in the follow-up examinations performed after three, six and twelve months.

As already described in the results of our previous study [16], both wound dressings appear to be subjectively equal in terms of patient satisfaction. However, the VSS [31,32,33] evaluations showed some subjective differences, consistent with the results of our previous study comparing Dressilk^®^ to Biobrane^®^ [6]. We were able to detect favorable results for Dressilk^®^ in the VSS, especially in terms of blood flow (month 6, *p* = 0.005). Similarly, as shown in another study, this was also found in the POSAS Observer scale [16].

#### 4.1.2. Objective Scar Assessment

Additionally, consistent with our previous studies [6,7,34], we evaluated the scars by using objective scar-assessment tools. As outlined in the literature, these tools are especially necessary to minimize inter-examiner variability and maximize the reliability of the results [35]. 

As a tool to detect elasticity, the Cutometer^®^ has already been validated as a reliable scar-assessment tool [36,37,38]. In this study, a significant difference in skin firmness (R0, *p* = 0.013) was observed only within three months. Interestingly, the scar previously treated with Dressilk^®^ did not show a difference in firmness and elasticity in comparison with normal skin from the three-month measurement onward. Although scar elasticity has been suggested to not reach the elasticity of normal skin, as shown in the measurements reported by Anthonissen et al. [39], this was not consistent with our findings. Nevertheless, it should be emphasized that scarring depends on the initial burn depth. Superficial burns usually heal without scarring, whereas the deep partial burns involving the dermal layer beneath the papillary layer cause scarring [40,41]. Moreover, the dermis containing the protein elastin might not be severely damaged in superficial-thickness burns, leading to our results. Interestingly, the data reported by Busche et al. suggest that even two years after superficial burn injury, the scars of superficial burn injuries solely treated with fatty gaze did not show elasticity similar to that of normal skin [29].

The Mexameter^®^ allows evaluation of skin vascularization (erythema) and pigmentation (melanin) [22]. External evaluation in previous studies has also been shown to be reliable in comparison with subjective scar-assessment tools such as the VSS [42,43,44]. Superficial partial-thickness burns are known to lead to alterations of skin pigmentation [34,45,46]. Inter-individual influences in factors such as sun or UV protection may also influence objective assessments. In our objective scar assessment, we showed that the wound areas were redder with higher erythema than normal skin after 12 months, consistent with the findings of a previous study [34]. Nonetheless, the pigmentation after Dressilk^®^ application seemed to be superior, showing no difference to normal skin at this point. In contrast, subjective scar assessments, as mentioned above and shown in the literature [47], can indicate non-apparent differences even sooner. Fortunately for burn surgeons, recent studies have shown that the highest patient satisfaction is related to coloration of the scar in conservatively treated superficial burn injuries [29]. For cases involving unsatisfactory hypopigmentation of burn scars, Busch et al. showed promising results by examining medical needling in combination with the ReCell Technique in 20 patients. In these cases, the melanin level increased significantly after 12 months while also showing subjective individual improvement of the scars [1].

The Tewameter^®^ is a modern digital evaporimeter that has been used in different studies and has been shown to be a reliable tool [23,25,48]. The TEWL is an essential parameter to determine the efficiency of the skin barrier [25]. Interestingly, we were able to show that, after six months, the initial wound area treated with Dressilk^®^ did not differ from normal skin, compared to the initial wound area treated with Suprathel^®^. Although this difference was not verifiable after 12 months, it may indicate slightly faster skin barrier normalization in wounds treated with Dressilk^®^. Consistent with this finding, the mean TEWL for Dressilk^®^ was lower than for Biobrane^®^ after 6 months in our previous study, although the difference was not statistically significant [34]. Even more, this finding indicates that the stratum corneum can restore its impaired function within months after the burn injury [25,39]. Consistent with our results, studies have shown that the primarily higher scar TEWL values gradually disappear over time [25]. However, some parameters can also influence these differences in TEWL, e.g., individual differences in skin treatments such as consequent moisturizing. We aimed to avoid the effects of these phenomena by choosing an intra-individual study design.

The O2C^®^ device assesses perfusion by a combination of photo-spectroscopy and Laser Doppler. Measurements of sO_2_, rHb and flow can be obtained using this approach [27]. This device has gained interest as a monitoring device after reconstructive surgery, especially for the evaluation of postoperative recovery and noninvasive perfusion control [27,49,50]. Moreover, the remaining blood flow can be used as a parameter to express the healing potential of burn-injured areas [51]. In this study, the scar tissue formerly treated with Suprathel^®^ showed both a significantly higher flow rate than that treated with Dressilk ^®^ after 3 months (*p* = 0.041) and a higher sO_2_ after 12 months (*p* = 0.004). Although these measurements may indicate a possible beneficial wound-healing potential for Suprathel^®^, Dressilk^®^, contrastingly, did not show any significant difference in oxygen saturation compared to the healthy skin from the six-month measurement onward. In contrast, our prior study results did not show significant differences between treated (Dressilk^®^ vs. Biobrane^®^) and non-treated areas after 6 and 12 months, regardless of the applied dressing [34]. The informative value of these parameters in scar evaluation remains unclear, since the blood flow and oxygen saturation are also related to the initial burn depth [51].

### 4.2. Limitations

In this context, the influences of various conditions that cannot be controlled must be reviewed critically. For example, we could not exclude the potential influences of differences in outside temperatures, increased perspiration, or very dry, brittle skin on the measurement results. Strong exposure to the sun due to insufficient sun protection or visits to a tanning bed could also have led to different long-term results. Nevertheless, by performing exact photo-documentation of the measurement points and ensuring that the assessments were performed by the same investigator and in the same examination room, we were able to ensure maximal accuracy in the comparisons.

## 5. Conclusions

Objective and repeatable scar-assessment tools allow quantification of the benefits of wound dressings in relation to scar quality. Both evaluated dressing materials showed similarly good results in the long-term objective and subjective scar assessments after 12 months. Nevertheless, Dressilk^®^ showed slight benefits in different categories and an overall faster return to the characteristics of non-injured skin areas. Since Dressilk^®^ has already been implemented in the SOC for superficial burns at our clinic, our study results underline that Dressilk^®^ remains an interesting alternative to Suprathel^®^ for the treatment of superficial partial-thickness burns. Further research on a larger variety of scars is needed for implementing and finding ideal objective scar-evaluation methods and facilitating repeatability.

## Figures and Tables

**Figure 1 jcm-11-02857-f001:**
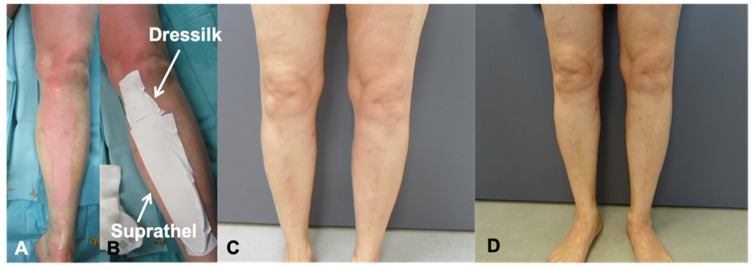
Partial thickness burn of the left leg; (**A**,**B**) before and after debridement; (**C**) 6-month follow-up; (**D**) 12-month follow-up.

**Figure 2 jcm-11-02857-f002:**
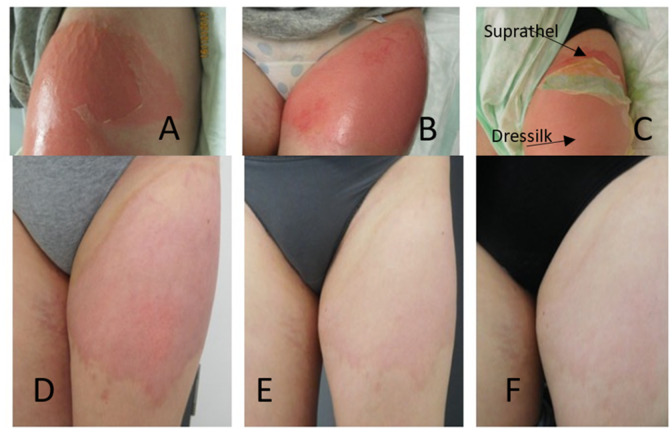
Partial thickness burn of the left leg; (**A**,**B**) before and after debridement; (**C**) application of Dressilk and Suprathel; (**D**) 3-month follow-up; (**E**) 6-month follow-up; (**F**) 12-month follow up.

**Figure 3 jcm-11-02857-f003:**
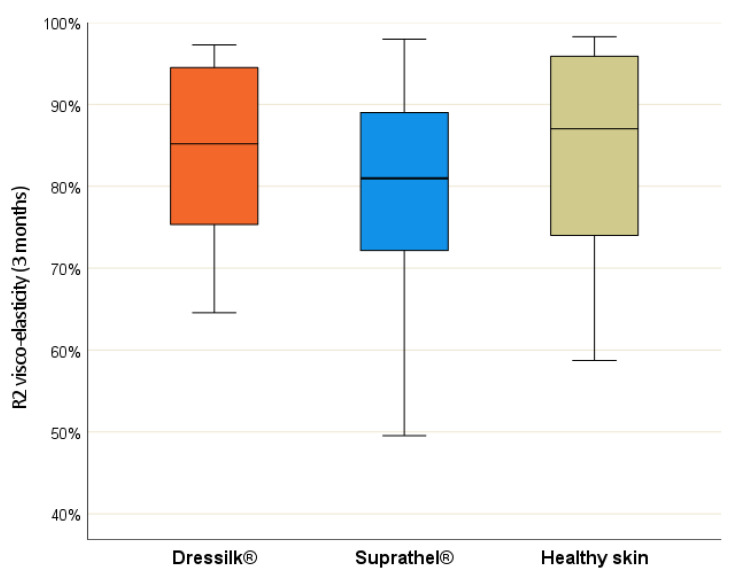
R_2_-values (visco-elasticity) after 3 months of areas treated initially with Suprathel (blue), Dressilk(orange) and the uninjured control area (green).

**Figure 4 jcm-11-02857-f004:**
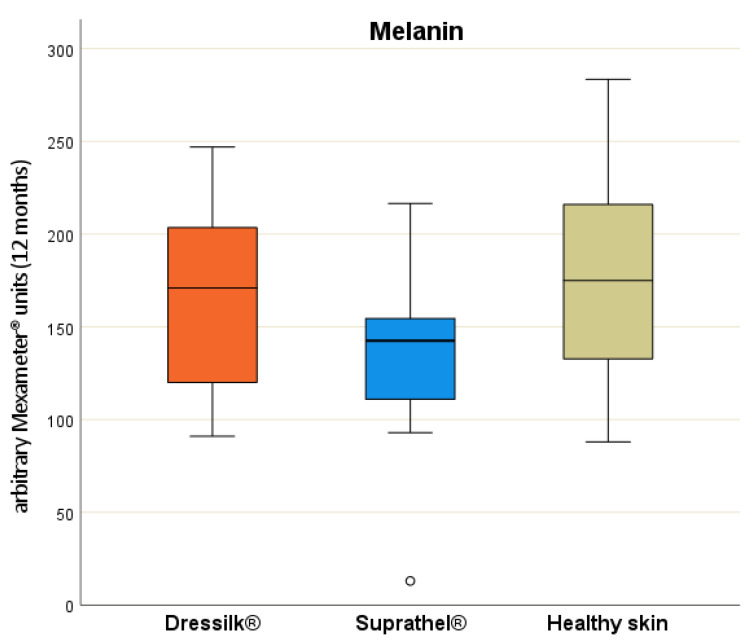
Level of melanin in arbitrary Mexameter^®^ units after 12 months of areas treated initially with Suprathel (blue), Dressilk (orange) and the uninjured control area (green).

**Figure 5 jcm-11-02857-f005:**
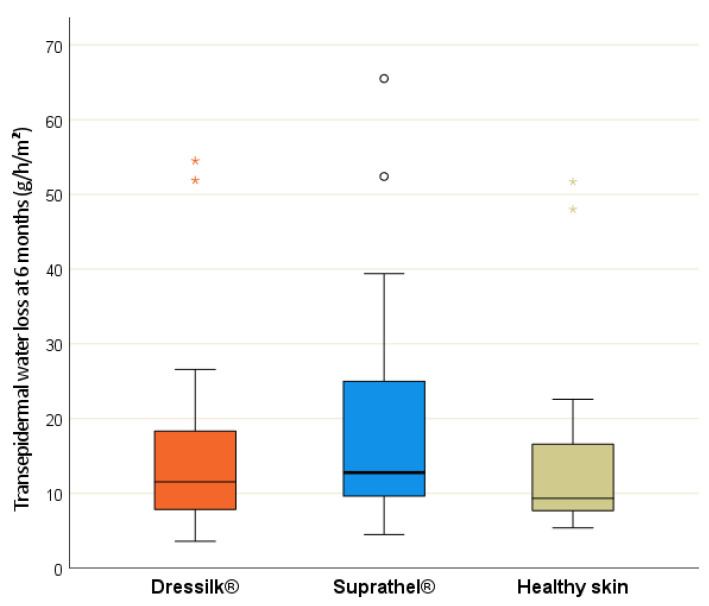
Transepidermal water loss (TEWL) in g/h/m^2^ after 6 months of areas treated initially with Suprathel (blue), Dressilk (orange) and the uninjured control area (green), significant differences marked.

**Figure 6 jcm-11-02857-f006:**
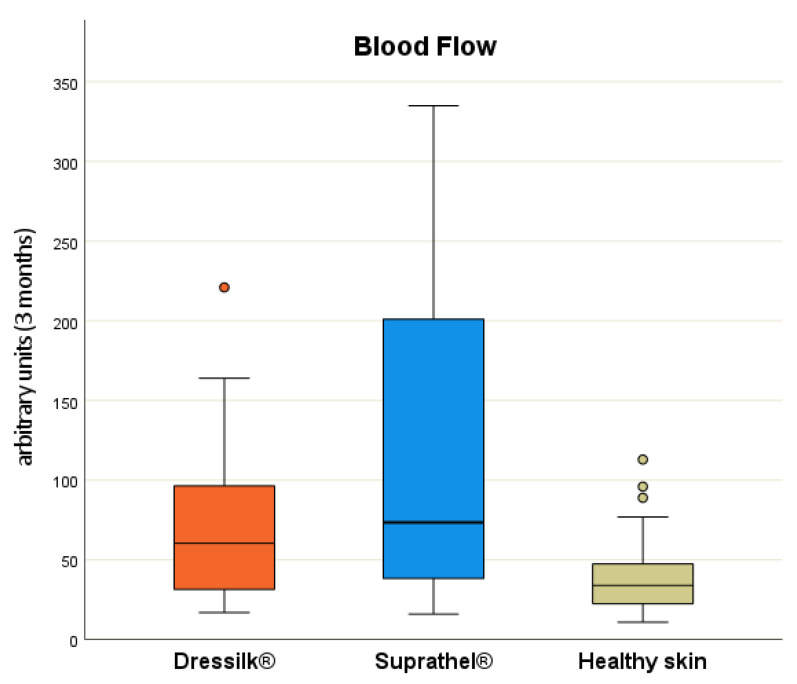
Flow in arbitrary units after 3 months of areas treated initially with Suprathel (blue), Dressilk (orange) and the uninjured control area (green), significant differences marked.

**Table 1 jcm-11-02857-t001:** Patient characteristics (age, sex and location of treated injury).

Patient ID	Sex	Age	Area Treated with Dresssilk®	Area Treated with Suprathel®
1	M	>40–50	Right forearm	Right hand
2	M	>60	Left thigh	Right thigh
3	M	>30–40	Left thigh	Right thigh
4	M	>50–60	Right hand and forearm	Right forearm
5	M	>40–50	Right forearm	Left forearm
6	M	>40–50	Left D1 + D2	Left D3–D5
7	F	>50–60	Right forearm	Left upper arm
8	F	>40–50	Left hand	Right hand
9	F	<20	Left thigh distal	Left thigh proximal
10	M	>30–40	Left upper arm	Left forearm
11	F	>20–30	Right thigh distal	Right thigh proximal
12	F	>40–50	Right breast	Abdomen
13	M	>30–40	Left hand and forearm	Right hand and forearm
14	M	>20–30	Right proximal forearm	Right hand and forearm
15	M	>20–30	Right hip	Right hand
16	M	>30–40	Left forearm proximal	Right forearm distal
17	M	>60	Right upper arm	Right forearm
18	F	>20–30	Abdomen	Abdomen
19	F	>50–60	Right thigh	Right shank
20	F	>50–60	Left foot and upper leg	Left shank and forearm

**Table 2 jcm-11-02857-t002:** Cutometer R_0_-values (stretchability/firmness) in mm and R_2_-values (visco-elasticity) in %, F1-values (elasticity) in mm^2^ after 3, 6 and 12 months of areas treated initially with Suprathel, Dressilk and the uninjured control area.

Cutometer-Measurement	Month	Dressilk^®^		Suprathel^®^		Healthy Skin	
		Mean	SD	Mean	SD	Mean	SD
R0	3	0.57	0.45	0.44	0.36	0.58	0.38
	6	0.83	0.54	0.79	0.53	0.79	0.59
	12	0.83	0.43	0.75	0.39	0.82	0.4
R2	3	84.02%	0.11	79.44%	0.13	84.87%	0.12
	6	85.63%	0.07	81.24%	0.12	83.18%	0.11
	12	82.32%	0.13	81.84%	0.11	82.36%	0.13
F1	3	0.09	0.08	0.07	0.06	0.07	0.05
	6	0.11	0.08	0.15	0.15	0.11	0.08
	12	0.17	0.12	0.15	0.07	0.16	0.09

**Table 3 jcm-11-02857-t003:** Level of melanin/erythema in arbitrary Mexameter^®^ units after 3, 6 and 12 months of areas treated initially with Suprathel, Dressilk and the uninjured control area.

Mexameter Measurements	Month	Dressilk^®^		Suprathel^®^		Healthy Skin	
		Mean	SD	Mean	SD	Mean	SD
Melanin	3	100	56	91	48	123	57
	6	118	43	106	58	138	53
	12	163	5	135	45	175	59
Erythema	3	429	11	441	141	310	120
	6	354	119	335	116	268	98
	12	288	97	280	114	237	100

**Table 4 jcm-11-02857-t004:** Transepidermal water loss (TEWL) in g/h/m^2^ after 3, 6 and 12 months of areas treated initially with Suprathel, Dressilk and the uninjured control area.

Tewameter Measurement	Month	Dressilk^®^		Suprathel^®^		Healthy Skin	
		Mean	SD	Mean	SD	Mean	SD
TEWL	3	18	11	22	13	13	9
	6	16	14	20	16	14	13
	12	15	11	15	10	12	7

**Table 5 jcm-11-02857-t005:** Oxygen levels in %, rHb/Flow in AU after 3, 6 and 12 months of areas treated initially with Suprathel, Dressilk and the uninjured control area.

O2C Measurements	Month	Dressilk^®^		Suprathel^®^		Healthy Skin	
		MW	SD	MW	SD	MW	SD
sO_2_	3	65	22	69	18	47	16
	6	56	22	62	22	50	17
	12	49	22	59	24	49	17
rHb	3	93	13	93	14	74	10
	6	87	14	87	13	76	12
	12	83	13	86	17	77	10
Flow	3	72	53	108	93	42	29
	6	79	53	73	57	48	35
	12	47	32	56	57	44	21

## Data Availability

Data can be obtained through the authors.

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
