# Peer review of "Evaluation of Scar Quality after Treatment of Superficial Burns with Dressilk^®^ and Suprathel^®^—In an Intraindividual Clinical Setting"

_jcm, 2022, doi:10.3390/jcm11102857_

Round 1
Reviewer 1 Report
Evaluation of scar quality after treatment of superficial burns with Dressilk and Suprathel – in an intraindividual clinical setting
In this study two dressings (Suprathel and Dressilk) were compared after superficial burn injuries using objective approaches to assess scar quality. This study is interesting as objective data on the performance of dressing materials are needed.
However, I have some major issues with the study in its current state.
Data from the same patient collective have been published before – in a study on patient comfort and subjective scar evaluation. The data presented here clearly are something different and are basically fine, however it should be mentioned in the text (e.g. in materials and methods with the patient characteristics in table 1) – especially since this table is basically the same as in Ref 16; Schiefer at al., Int Wound J 2022.
Obviously, a copy/paste error occurred in figure 1: it is NOT a right forearm and hand here (that was the case in the REF16 paper).
In my opinion, the presentation of the results should be substantially improved. Tables are partially redundant to the graphs; significance levels are missing in the graphs, some graphs can hardly be read, figure legends should contain more information on the data shown, not only a heading. Throughout the results and the discussion it is difficult to follow whether there is a difference between the two dressings or between the dressings and “normal” healthy skin.
Moreover, I have to ask, about the data for example in table 2: cutometer R0 data: the SD with 50-90% of the mean is pretty high – is that normal? And, the authors should think about how many digits are necessary: e.g. Table 4: mean = 17,57 SD = 10,90 – in my opinion no digits would be necessary. The layout of tables and graphs could be improved in order to facilitate reading/understanding the basically interesting data.
For oxygen saturation, the authors should take care not to abbreviate with SO2 (which would stand for sulfur dioxide…).
Author Response
Dear reviewer,
your hard work is much appreciated and I wish to address each of your comments to the best of my knowledge.
In this study two dressings (Suprathel and Dressilk) were compared after superficial burn injuries using objective approaches to assess scar quality. This study is interesting as objective data on the performance of dressing materials are needed.
However, I have some major issues with the study in its current state.
Data from the same patient collective have been published before – in a study on patient comfort and subjective scar evaluation. The data presented here clearly are something different and are basically fine, however it should be mentioned in the text (e.g. in materials and methods with the patient characteristics in table 1) – especially since this table is basically the same as in Ref 16; Schiefer at al., Int Wound J 2022.
This is a good point, we have already mentioned this in the introduction section.
“Since many burn centers prefer using Suprathel® for these wounds, we had previously compared wound healing and patient satisfaction after treatment of superficial burns with Dressilk® and Suprathel® [16]“ but now also added the reference to the methods section.
Obviously, a copy/paste error occurred in figure 1: it is NOT a right forearm and hand here (that was the case in the REF16 paper).
We apologize this mistake and corrected it in the figure legend.
In my opinion, the presentation of the results should be substantially improved. Tables are partially redundant to the graphs; significance levels are missing in the graphs, some graphs can hardly be read, figure legends should contain more information on the data shown, not only a heading. Throughout the results and the discussion it is difficult to follow whether there is a difference between the two dressings or between the dressings and “normal” healthy skin.
We removed some graphs since the information can be found in the tables. Additionally, we updated the figure legends and added underlined some sentences for better understanding, especially regarding the difference between the two dressings after 12 months.
Moreover, I have to ask, about the data for example in table 2: cutometer R0 data: the SD with 50-90% of the mean is pretty high – is that normal? And, the authors should think about how many digits are necessary: e.g. Table 4: mean = 17,57 SD = 10,90 – in my opinion no digits would be necessary. The layout of tables and graphs could be improved in order to facilitate reading/understanding the basically interesting data.
Thank you. We removed the digits (except in the first table) and tried to change the tables for better readability. Since patients have a different age, different burn locations and different skin pigmentation high SD are possible.
For oxygen saturation, the authors should take care not to abbreviate with SO2 (which would stand for sulfur dioxide…).
This is a very good point, we changed it in the manuscript.
Reviewer 2 Report
Over the last years the long-term consequences of scar quality are interesting topics in plastic surgery and this article analyse all aspect of this issue. The manuscript is clear, relevant for the field and presented in a well-structured manner with the mostly recent publications.
The manuscript is scientifically sound and is the experimental design is appropriate to test the hypothesis
The tables/images/schemes are appropriate and the date are easy to interpret and understand but there are photos of only one patient. In my opinion it is necessary to include al least another clinical case with its images.
The data interpreted appropriately and consistently throughout the manuscript.
The conclusions are consistent with the evidence and arguments presented
Author Response
Dear reviewer,
Your hard work is much appreciated and I wish to address each of your comments to the best of my knowledge.
Thank you very much for your comments, as suggested we added a further clinical case with photos.
Round 2
Reviewer 1 Report
The manuscript has improved to some extent - however I find it is a pity, that basically all the graphs visualizing the complex results were removed, even of data are summarized in tables.
In my opinion, just underlining a sentence does not make it easier to read/understand it. The results section would clearly benefit from some real rewriting regarding consistency. Moreover, I doubt that underlining senetences is allowed in the publication/layout specifications.
Author Response
Dear reviewer,
thank you very much for your work.
We changed the underlined results and instead now added a short summery of all significant results at the end of the results section for better understanding.
Additionally, we again added some figures in better quality then last time and with expanded subtitles.